# Imaging the coupling between itinerant electrons and localised moments in the centrosymmetric skyrmion magnet GdRu$_2$Si$_2$

Yuuki Yasui [1✉], Christopher J. Butler [1✉], Nguyen Duy Khanh[1,8], Satoru Hayami[2,3], Takuya Nomoto[2], Tetsuo Hanaguri [1✉], Yukitoshi Motome [2], Ryotaro Arita[1,2], Taka-hisa Arima[1,4], Yoshinori Tokura [1,2,5] & Shinichiro Seki[2,6,7]

Magnetic skyrmions were thought to be stabilised only in inversion-symmetry breaking structures, but skyrmion lattices were recently discovered in inversion symmetric Gd-based compounds, spurring questions of the stabilisation mechanism. A natural consequence of a recent theoretical proposal, a coupling between itinerant electrons and localised magnetic moments, is that the skyrmions are amenable to detection using even non-magnetic probes such as spectroscopic-imaging scanning tunnelling microscopy (SI-STM). Here SI-STM observations of GdRu$_2$Si$_2$ reveal patterns in the local density of states that indeed vary with the underlying magnetic structures. These patterns are qualitatively reproduced by model calculations which assume exchange coupling between itinerant electrons and localised moments. These findings provide a clue to understand the skyrmion formation mechanism in GdRu$_2$Si$_2$.

[1] RIKEN Center for Emergent Matter Science, Wako, Saitama 351-0198, Japan. [2] Department of Applied Physics, The University of Tokyo, Bunkyo, Tokyo 113-8656, Japan. [3] Department of Physics, Hokkaido University, Sapporo, Hokkaido 060-0810, Japan. [4] Department of Advanced Materials Science, The University of Tokyo, Kashiwa, Chiba 277-8561, Japan. [5] Tokyo College, The University of Tokyo, Bunkyo, Tokyo 113-8656, Japan. [6] PRESTO, Japan Science and Technology Agency (JST), Kawaguchi 332-0012, Japan. [7] Institute of Engineering Innovation, The University of Tokyo, Bunkyo, Tokyo 113-8656, Japan. [8] Present address: Institute for Solid State Physics, The University of Tokyo, Chiba 277-8581, Japan. ✉email: yuuki.yasui@riken.jp; christopher.butler@riken.jp; hanaguri@riken.jp

Magnetic skyrmions are topologically protected swirling spin structures which have been observed in inversion-symmetry breaking structures, in which they are stabilised by the Dzyaloshinskii–Moriya (DM) interaction[1–6]. Recently discovered skyrmion lattices in inversion symmetric crystals[7–9] have been proposed to be stabilised instead by geometrical frustration[10,11], or by multiple-spin interactions involving itinerant electrons[12,13,14–17]. The latter mechanism can be expected to apply in $GdRu_2Si_2$, as it has a tetragonal structure belonging to the space group $I4/mmm$ (Fig. 1a), in which geometrical frustration should be absent. The multiple-spin interactions have been theoretically argued to be mediated by itinerant electrons[14,15], but experimental support is so far lacking.

$GdRu_2Si_2$ hosts a variety of magnetic orders[18–20], whose localised moments are provided by Gd $4f^7$ orbitals, and its itinerant electrons mostly come from Ru $4d$ orbitals with minor contributions from Si $3p$ and Gd $5d$ orbitals[21]. Very recently, resonant X-ray scattering (RXS) and Lorentz transmission electron microscopy experiments have revealed the details of the magnetic structures of the Gd moments, including the square skyrmion lattice, under magnetic field applied parallel to the $c$-axis[9] (Fig. 1b). In this compound, the magnetic modulation vector $\mathbf{Q}_{mag} = (0.22, 0, 0)$ is observed to be common to all magnetic phases. At low magnetic field (Phase I), a screw-like spin texture is realised. In a narrow range between 2.1 and 2.6 T (Phase II), the double-$Q$ square skyrmion lattice is stabilised, where the magnetic structure can be approximately described by the superposition of two screw spin structures with orthogonally arranged magnetic modulation vectors. At higher magnetic field (Phase III), a fan structure has been proposed while it has not been concluded whether this Phase III is a single-$Q$ or double-$Q$ state. Magnetic moments are fully polarised (FP) above around 10 T (FP phase).

When itinerant electrons are involved in the formation of the magnetic orders, the relevant coupling between the itinerant electrons and localised magnetic moments may enable the detection of information about the magnetic structure through the charge channel. To experimentally verify and gain insight into such coupling, we performed spectroscopic-imaging scanning tunnelling microscopy (SI-STM) measurements on $GdRu_2Si_2$. These revealed that the local density of states (LDOS) forms characteristic spatial patterns that vary in accordance with magnetic structures, evidencing the intimate coupling between itinerant electrons and localised magnetic moments. The observed LDOS patterns clarify that not only Phase II but also Phase III hosts a double-$Q$ structure. These patterns are reasonably reproduced by a model calculation which assumes exchange coupling between itinerant electrons and localised magnetic moments.

## Results

We inspected multiple cleaved surfaces and observed two types of termination as shown in Fig. 1c, d. One of the terminations showed a clear atomic lattice with the lattice constant corresponding to either Gd-Gd or Si-Si in-plane distance (Fig. 1c). This suggested that cleavage occured between Gd and Si layers. Atomic corrugations were hardly seen on the other termination surface even with the identical scanning tip (Fig. 1d). Among seven samples we investigated, we did not observe any surface with atomic corrugations corresponding to Ru–Ru lattice spacing.

Figure 1e, f shows that two types of surfaces exhibit different tunnelling conductance $dI/dV$ spectra, which are taken at surfaces shown in Fig. 1c, d, respectively. Here, $I$ is the tunnelling current, and $V$ is the sample bias voltage. To identify the termination, the observed $dI/dV$ spectra, which are proportional to the LDOS at a given tip height, are compared with first-principle calculations. The calculations are performed based on the density functional theory (DFT) for slab systems, where we assume collinear ferromagnetic order. The overall correspondence allows us to assign the termination of Fig. 1c to Si, and that of Fig. 1d to Gd. Hereafter, we will discuss the Si-terminated surface since the atomic and electronic modulations are more clearly observed for this surface. Additionally, the Gd-terminated surface shows properties different from those expected from the bulk behaviour, which may be induced by surface effects (see Supplementary Note 1).

To investigate the impact of the magnetic order on the charge channel, SI-STM is performed in a magnetic field range that covers all of the magnetic phases at low temperature. For the spectroscopic imaging, full $I(V)$ and $\frac{dI(V)}{dV}$ curves were recorded at

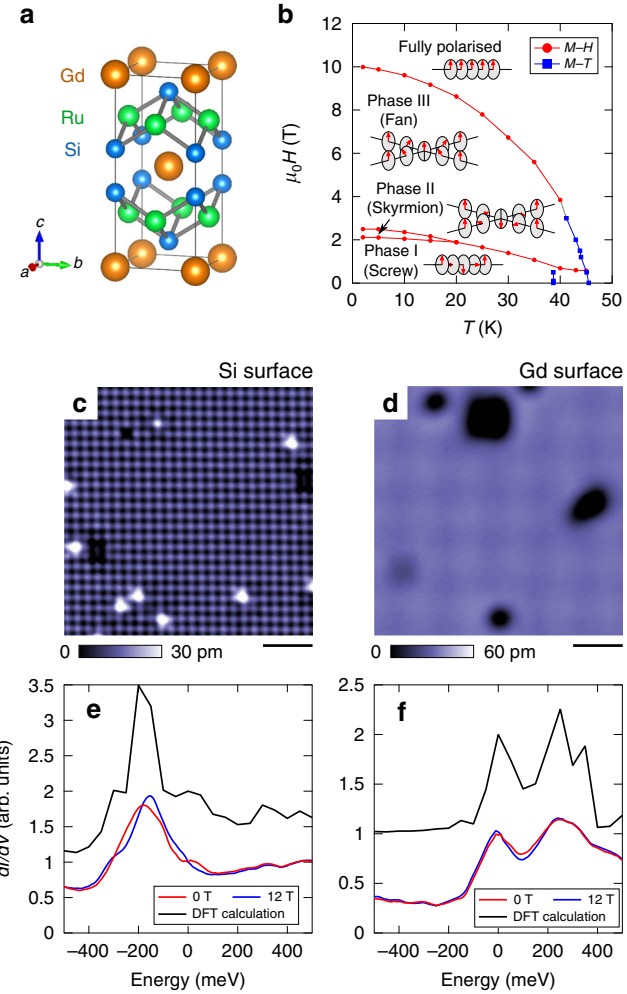

**Fig. 1 Basic properties of GdRu₂Si₂. a** Crystal structure of $GdRu_2Si_2$. **b** Magnetic phase diagram for magnetic field $H$ applied parallel to the $c$-axis. Data points shown with red circles and blue squares were obtained using $H$ and $T$ dependence of the magnetisation, respectively[9]. Schematics of the magnetic structures are depicted in insets. **c, d** Constant-current topographic images of Si-terminated and Gd-terminated surfaces, respectively, at $T = 1.5$ K and $\mu_0 H = 0$ T. The setup sample bias voltage was $V_s = 100$ mV, and tunnelling current was $I_s = 100$ pA. Scale bar 2 nm. **e, f** Spatially averaged conductance spectra for Si-terminated and Gd-terminated surfaces, respectively. The setup condition is $V_s = 500$ mV and $I_s = 100$ pA. The bias modulation amplitude was $V_{mod} = 10$ mV. Calculated LDOS spectra for collinear ferromagnetic order at a tip position 5 Å above the surface are also shown and vertically shifted for clarity.

**Fig. 2 SI-STM results on the Si-terminated surface at $T$=1.5 K and $\mu_0H$ = 2.3 T in Phase II (skyrmion). a** Constant-current topographic image. The setup condition is $V_s$ = 100 mV and $I_s$ = 100 pA. Scale bar 5 nm. **b** Normalised conductance map $L(\mathbf{r}, E = -20$ meV) taken in the same field of view. $V_s$ = 100 mV, $I_s$ = 100 pA, and $V_{mod}$ = 5 mV. Scale bar 5 nm. **c** Fourier transform (FT) of (**b**). Peaks appear at $\mathbf{Q}_1$ = (0.22, 0) and $\mathbf{Q}_2$ = (0, 0.22) (yellow circles), $\mathbf{Q}_1 \pm \mathbf{Q}_2$ (blue squares), $2\mathbf{Q}_1$ and $2\mathbf{Q}_2$ (orange diamonds), and atomic Bragg points $\mathbf{G} \equiv$ (1, 0) and (0, 1) (red triangles). Scale bar 1 nm$^{-1}$.

each pixel to obtain spatial dependence and bias dependence simultaneously. We analyse normalised conductance maps $L(\mathbf{r}, E = eV) \equiv \frac{dI(\mathbf{r},V)}{dV} / \frac{I(\mathbf{r},V)}{V}$ instead of raw conductance maps $\frac{dI(\mathbf{r},V)}{dV}$ to suppress artefacts from the constant-current feedback loop[22,23]. Here, $\mathbf{r}$ is the lateral position, and $e$ is the elementary charge. We begin our discussion from Phase II, which is identified as the square skyrmion lattice phase. Figure 2a, b shows a constant-current topograph and a $L(\mathbf{r}, E = -20$ meV) map in the same field of view, respectively. In the $L(\mathbf{r}, E)$ map, a four-fold symmetric superstructure with a period of 1.9 nm is observed. This period corresponds to that of the skyrmion lattice previously determined using RXS[9]. Therefore, we infer that the pattern of the skyrmion lattice is imprinted in the LDOS of itinerant electrons. As shown in Fig. 2c, Fourier analysis clarifies periodic components in the $L(\mathbf{r}, E)$ map. In addition to atomic Bragg peaks at $\mathbf{G} \equiv$ (1, 0) and (0, 1), several modulation vectors $\mathbf{Q}$ are observed. Modulations with the smallest $|\mathbf{Q}|$ are found at $\mathbf{Q}_1$ = (0.22, 0) and $\mathbf{Q}_2$ = (0, 0.22). We also observed peaks at $\mathbf{Q}_1 \pm \mathbf{Q}_2$, $2\mathbf{Q}_1$, and $2\mathbf{Q}_2$. Other peaks are assigned to 'replicas' of these Q-vectors shifted by $\mathbf{G}$. (see Supplementary Fig. 3 for higher spatial resolution data, Supplementary Fig. 4 for bias voltage dependence, and Supplementary Fig. 5 for location dependence of the spectra.)

LDOS patterns in the other magnetic phases are also investigated. The electronic modulations clearly change depending on the magnetic phase, as seen in $L(\mathbf{r}, E = -20$meV) maps (Fig. 3a–d) and their Fourier transformed images (Fig. 3e–h). (see Supplementary Fig. 6 for raw $dI/dV$ maps at different magnetic fields and Supplementary Fig. 7 for the data indicating the robustness of the tip throughout the measurements.) In Phase I, the LDOS forms a two-fold symmetric pattern, which is composed of modulation vectors $2\mathbf{Q}_1$ and $\mathbf{Q}_1 + \mathbf{Q}_2$. In Phase III, the LDOS pattern is four-fold symmetric and is characterised by $2\mathbf{Q}_1$ and $2\mathbf{Q}_2$. While the previous spatial-averaging RXS experiments cannot distinguish a double-$Q$ order and a multiple-domain state of single-$Q$ order, the present real-space imaging clarifies a double-$Q$ order is realised in Phase III. In the FP phase, all Q-vectors disappear except for atomic Bragg peaks. It should be noted that $\mathbf{Q}_1$ and $\mathbf{Q}_2$ modulations are observed only in Phase II. We discuss this point below.

To further corroborate the correspondence between the LDOS and magnetic orders, we investigate detailed magnetic-field dependence of the $dI/dV$ spectrum. Figure 3i shows a series of spatially averaged $dI/dV$ spectra for $\mu_0H$≤3 T. The spectrum varies only subtly within each magnetic phase. On the other hand, it exhibits discontinuous changes across phase boundaries at 2.275 and 2.6 T. Note that these transition fields slightly change depending on the field history. Such first-order-like transitions

are consistent with the magnetic measurement[9]. They also reflect the transition between topologically trivial and non-trivial phases. Above 3 T, as shown in Fig. 3j, the spectrum evolves continuously until it saturates in the FP phase above 10 T. The trend is clearly seen by plotting magnetic field dependence of $dI/dV$ at a chosen energy $E = -70$ meV (Fig. 3k). (see Supplementary Fig. 8 for the $dI/dV$ evolution at different energies, Supplementary Fig. 9 for the data taken with decreasing field, and Supplementary Fig. 10 for the same analyses for the normalised conductance.) The observed one-to-one correspondence between the LDOS and the magnetic phase indicates that itinerant electrons and localised magnetic moments are intimately coupled.

Let us compare the periods of observed LDOS modulations with previously reported magnetic structures[9]. In the LDOS maps, fundamental modulations of $\mathbf{Q}_1$ and $\mathbf{Q}_2$ appear only in Phase II while $2\mathbf{Q}_1$ and/or $2\mathbf{Q}_2$ show up in all the magnetic phases except for the FP phase. Namely, the LDOS takes on the period of the magnetic structure in Phase II; the LDOS period becomes a half of the magnetic period in Phase I and III. One may expect halved charge period in systems with coupled charge- and spin-density waves, where itinerant electrons host both spin and charge modulations[24]. However, such a simple relation in periods does not apply for Phase II (skyrmion lattice) of GdRu$_2$Si$_2$. The absence of $\mathbf{Q}_1$ modulation in Phase I ensures that the scanning tip is not magnetised due to unintentional pick-up of magnetic Gd atoms.

## Discussion

In order to understand the origin of the observed LDOS modulations, we performed calculations for magnetic configurations and charge-density distributions. The magnetic configurations are obtained for an effective spin model with long-range interactions that can be originated from the coupling between the itinerant-electron spins and localised spins. The Hamiltonian is given as:[15]

$$\mathcal{H} = 2\sum_\nu \left[ -J \left( \sum_{\alpha=x,y,z} \Gamma^{\alpha\alpha}_{\mathbf{Q}_\nu} S^\alpha_{\mathbf{Q}_\nu} S^\alpha_{-\mathbf{Q}_\nu} \right) + \frac{K}{N} \left( \sum_{\alpha=x,y,z} \Gamma^{\alpha\alpha}_{\mathbf{Q}_\nu} S^\alpha_{\mathbf{Q}_\nu} S^\alpha_{-\mathbf{Q}_\nu} \right)^2 \right] - H\sum_i S^z_i, \quad (1)$$

where $\mathbf{S}_{\mathbf{Q}_\nu} = (S^x_{\mathbf{Q}_\nu}, S^y_{\mathbf{Q}_\nu}, S^z_{\mathbf{Q}_\nu})$ is the Fourier transform of the localised spin $\mathbf{S}_i$ treated as a classical vector with the normalisation $|\mathbf{S}_i| = 1$, and $N$ is the system size. The Hamiltonian includes two exchange terms defined in momentum space: the bilinear exchange interaction $J$ and the biquadratic exchange interaction $K$. The wave numbers $\mathbf{Q}_\nu$ are set to be $\mathbf{Q}_1 = (\pi/3, 0)$ and $\mathbf{Q}_2 = (0, \pi/3)$. We also introduce an anisotropy due to the symmetry of the tetragonal crystal structure as $\Gamma^{yy}_{\mathbf{Q}_1} = \Gamma^{xx}_{\mathbf{Q}_2} = \gamma_1$, $\Gamma^{xx}_{\mathbf{Q}_1} = \Gamma^{yy}_{\mathbf{Q}_2} = \gamma_2$, and $\Gamma^{zz}_{\mathbf{Q}_1} = \Gamma^{zz}_{\mathbf{Q}_2} = \gamma_3$, which selects the spiral plane. The last term in Eq. (1) represents the Zeeman coupling to an external magnetic field $H$. Performing the simulated annealing

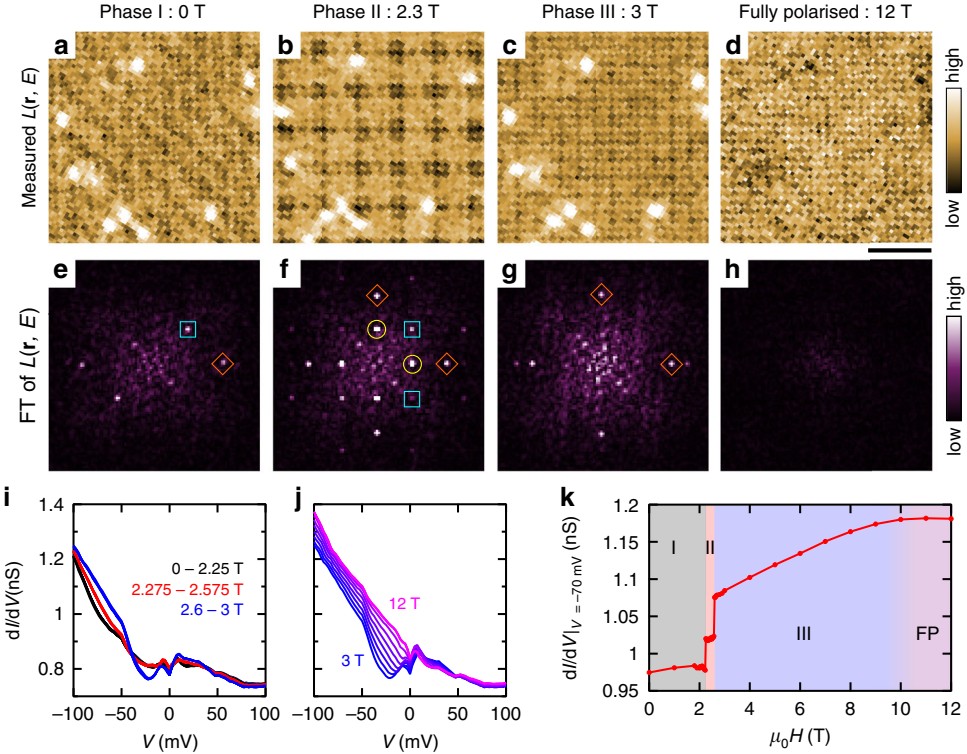

**Fig. 3 SI-STM for different magnetic phases. a–d** Normalised conductance maps $L(\mathbf{r}, E = -20\,\text{meV})$ in different magnetic phases measured at 1.5 K. The images were cropped from 30 × 30 nm²-large maps. $V_s = 100$ mV, $I_s = 100$ pA, and $V_{mod} = 5$ mV. Scale bar 3 nm. **e–h**, FTs of $L(\mathbf{r}, E = -20\text{meV})$ maps. **i, j** Spatially averaged d$I$/d$V$ curves (**i**) below 3 T and (**j**) above 3 T. A spectrum was taken every 25 mV between 2 and 2.7 T. $V_s = 100$ mV, $I_s = 100$ pA, and $V_{mod} = 2.5$ mV. **k** Magnetic-field dependence of the value of d$I$/d$V$ at $V = -70$ mV.

by means of Monte Carlo simulations for the $N = 96^2$ sites at $J = 1$, $K = 0.5$, $\gamma_1 = 0.9$, $\gamma_2 = 0.72$, and $\gamma_3 = 1$, we obtained the screw, skyrmion lattice, fan, and fully polarised states while increasing $H$. We show the spin configurations for each phase at $H = 0$, 0.6, 0.725, and $\infty$ in Fig. 4a, 4b, 4c, and 4d, respectively.

The square skyrmion lattice (Phase II) is found to be stabilised with the help of the anisotropy for the tetragonal crystal structure. Note that square skyrmion lattices in square crystal structures were not expected to be stabilised in previous reports, in which the magnetic anisotropy was not considered[15]. At a higher magnetic field, the calculation predicts a double-$Q$ fan structure in Phase III, consistent with the experiments (Fig. 3c, g).

The charge density is calculated by considering itinerant electrons coupled with the spin textures obtained as above. The Hamiltonian is given as:

$$\mathcal{H} = -t\sum_{\langle i,j\rangle,\sigma}(c^{\dagger}_{i\sigma}c_{j\sigma} + \text{h.c.}) + J_K\sum_i \mathbf{s}_i \cdot \mathbf{S}_i, \qquad (2)$$

where $c^{\dagger}_{i\sigma}$ ($c_{i\sigma}$) is the creation (annihilation) operator of an itinerant electron at site $i$ and with spin $\sigma$. The first term represents the nearest-neighbour hopping of electrons. The second term represents the spin–charge coupling between the electron spin $\mathbf{s}_i = (1/2)\sum_{\sigma,\sigma'}c^{\dagger}_{i\sigma}\boldsymbol{\sigma}_{\sigma\sigma'}c_{i\sigma'}$ and the underlying spin texture; $\boldsymbol{\sigma}$ denotes the Pauli matrix. We set $t = J_K = 1$ and the chemical potential $\mu = -3$. The charge density at site $i$, $\langle n_i\rangle = \langle\sum_{\sigma}c^{\dagger}_{i\sigma}c_{i\sigma}\rangle$, is obtained by diagonalising the Hamiltonian in Eq. (2) for each spin texture. The results and their Fourier transforms are shown in Fig. 4e–l. (see Supplementary Fig. 11 for the results with different chemical potentials.)

By comparing Fig. 4a–d and Fig. 4e–h, it can be seen that charge-distribution patterns reflect the magnetic structures. This

can be interpreted as follows. Since the itinerant electrons' spins are aligned with localised moments, kinetic energy of the itinerant electrons depends on the relative angle between localised magnetic moments at neighbouring sites. Thus, itinerant electrons reflect local magnetic structures. The charge modulations on the magnetic textures are qualitatively understood from the scattering process via the spin–charge coupling $J_K$. Within the second-order perturbation theory, the charge density at momentum $\mathbf{q}$ is proportional to $J_K^2\sum_{\mathbf{q}_1\mathbf{q}_2}\Lambda_{\mathbf{q}_1\mathbf{q}_2}(\mathbf{S}_{\mathbf{q}_1}\cdot\mathbf{S}_{\mathbf{q}_2})\delta_{\mathbf{q},\mathbf{q}_1+\mathbf{q}_2}$, where $\Lambda_{\mathbf{q}_1\mathbf{q}_2}$ is a form factor depending on the electronic structure and $\delta$ is the Kronecker delta. The nonzero $\mathbf{S}_{\mathbf{q}}$ components in each magnetic texture satisfying $\mathbf{S}_{\mathbf{q}_1}\cdot\mathbf{S}_{\mathbf{q}_2}\neq 0$ explain the wave numbers $\mathbf{q}$ for the charge modulations.

The calculated charge modulations resemble the basic features of the observed LDOS structures. The wavy modulation orthogonal to the screw structure in Phase I results in the stripe pattern in charge density (Fig. 4e). $2\mathbf{Q}_1$ and $2\mathbf{Q}_2$ appear in all the magnetic phases except for the FP phase and dominate in Phase I and III (Fig. 4i–l). This is because the local configuration of relative angles between neighbouring spins becomes almost the same every half periodicity of the magnetic modulations. In contrast, in Phase II, the angles between neighbouring spins at the skyrmion core and in between the cores are different. Therefore, $\mathbf{Q}_1$ and $\mathbf{Q}_2$ modulations appear in the charge sector. It should be noted that the peak at $\mathbf{Q}_1 + \mathbf{Q}_2$ in Phase I cannot be explained by the present model, and more advanced model may be necessary to explain this behaviour. Nevertheless, the overall good agreement between the observed and calculated spatial patterns in the double-$Q$ states suggests that the present theoretical framework based on multiple-spin interactions well captures the physics behind the skyrmion formation in this centrosymmetric magnet.

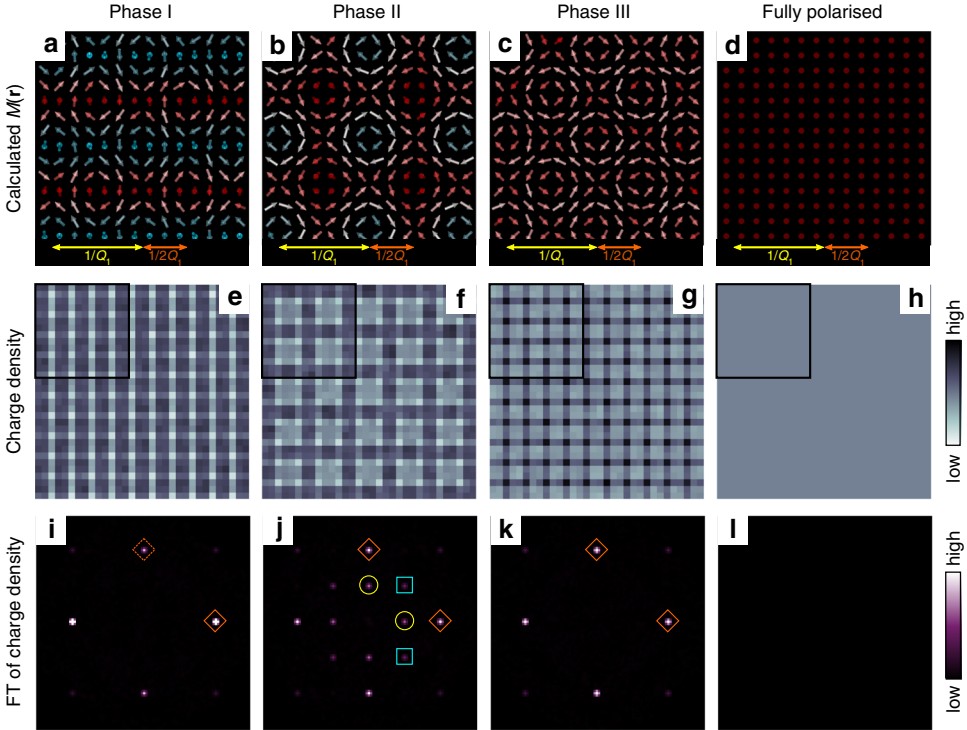

**Fig. 4 Magnetic and electronic structure by the model calculation. a–d** Calculated local magnetic moments $M(\mathbf{r})$ in different magnetic phases. Red (cyan) colour represents the magnetic moments pointing (anti)parallel to the magnetic field. **e–h** Calculated charge-density patterns in different magnetic phases. The regions shown in (**a**)–(**d**) are marked with black squares. **i–l** FTs of (**e**)–(**h**). Coloured symbols correspond to those in Fig. 2.

We note that magnetic structures, including skyrmions, have also been detected with non-magnetic STM tips via the mechanisms known as the tunnelling anisotropic magnetoresistance (TAMR)[3,25–28] and the non-colinear magnetoresistance (NCMR)[28,29] in $3d$ transition metals where the magnetic structures are originated from magnetic moments carried by itinerant electrons. The TAMR effect may not explain the present observation. This is because the centre and the edges of skyrmions show different contrast in the present LDOS map, whereas spins pointing in and out of the surface should appear similarly for the TAMR effect. By contrast, the observed LDOS modulations are similar to those caused by the NCMR. In the case of GdRu$_2$Si$_2$; however, the NCMR effect alone is not enough to explain the present observation because the coupling between itinerant electrons and localised moments is indispensable. Our observations evidence such a coupling, which not only allows us to access the localised moments from the charge sector but also may play a role for the itinerant-electron mediated magnetic interactions responsible for the skyrmion formation.

In conclusion, our observation of modulations of itinerant electrons associated with magnetic structures provides evidence for a coupling between itinerant-electron states and local magnetic moments in the centrosymmetric skyrmion magnet GdRu$_2$Si$_2$. The observed modulations are reproduced by charge density calculations which consider exchange coupling between itinerant electrons and localised magnetic moments fixed by anisotropic multiple-spin interactions. We interpret that this happens because spatially varying kinetic energy of itinerant electrons reflects neighbouring configurations of Gd moments. These results together have established the basic framework of the coupling between itinerant electrons and local magnetic moments in GdRu$_2$Si$_2$. Further theoretical and experimental investigation is required to explain the detailed features in the observed modulations (such as $\mathbf{Q}_1 + \mathbf{Q}_2$ component in Phase I), which may also

lead to identify the microscopic formation mechanism of the square skyrmion lattice in the absence of the DM interaction.

## Methods

**Sample preparation and STM measurements**. GdRu$_2$Si$_2$ single crystals were grown with the floating zone method[9]. The samples were cleaved in an ultra-high vacuum chamber ($\sim10^{-10}$ Torr) at around 77 K to expose clean and flat (001) surfaces and then transferred to the microscope[30] without breaking vacuum. As scanning tips, tungsten wires were used after electro-chemical etching in KOH aqueous solution, followed by tuning using field ion microscopy and controlled indentation at clean Cu(111) surfaces. All the measurements were conducted at temperature $T \simeq 1.5$ K, and magnetic field was applied along the crystalline $c$-axis. Tunnelling conductance was measured using the standard lock-in technique with AC frequency of 617.3 Hz.

**Calculation of the density of states**. The local density of states shown in the main text are obtained from first principles calculations for slab systems. The actual calculations are performed based on DFT with VASP code[31,32], where we assume a collinear ferromagnetic order. We consider the conventional cell of GdRu$_2$Si$_2$ with the experimental lattice parameters[33], $a = 4.1634$ Å, $c = 9.6102$ Å, and $z_{Si} = 0.375$, and then, stack it to construct the supercell systems with eight Ru-layers. Finally, we insert a vacuum layer with 10 Å at the edge of the slabs, and perform a surface relaxation calculation to optimise the positions of surface atoms. The LDOS spectra are calculated as the summation of partial charge densities of the Bloch states, $\sum' |\psi_{n\mathbf{k}}(\mathbf{r})|^2$, where the summation $\sum'$ is restricted to $(n\mathbf{k})$ with the energy $\varepsilon_{n\mathbf{k}} \in [\varepsilon - \Delta, \varepsilon + \Delta]$. We employ the exchange-correlation functional proposed by Perdew et al.[34], $E_c = 450$ eV as the cutoff energy for the planewave basis set, and $N_\mathbf{k} = 10 \times 10 \times 1$ as the number of $\mathbf{k}$-points for the self-consistent calculation. In the LDOS calculations, we use a denser k-mesh, $N_\mathbf{k} = 40 \times 40 \times 1$ and $\Delta = 25$ meV.

## Data availability

The data that support the findings of this study are available from the corresponding author upon reasonable request. SI-STM data for Gd-terminated surface and additional data supporting the main observation are presented in the Supplementary Information.

## Code availability

The codes used for this study are available from the corresponding author upon reasonable request.

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

## Acknowledgements

The authors acknowledge M. Hirschberger, K. Ishizaka, Y. Kohsaka, T. Machida, and Y. Ohigashi for discussion. This work was supported by JST CREST Grant Nos. JPMJCR16F2, JPMJCR18T2, and JPMJCR1874, by Grant-in-Aid JSPS KAKENHI Grant Nos. JP19H05824, JP19H05825, JP19H05826, JP18K13488, JP20H00349, and JP18H03685, by JST PRESTO Grant No. JPMJPR18L5, and by Asahi Glass Foundation. C.J.B. acknowledges support from RIKEN's SPDR fellowship.

## Author contributions

T.H., T.-h.A., Y.T., and S.S. conceived the project. N.D.K. synthesised GdRu$_2$Si$_2$ single crystals. Y.Y., C.J.B., and T.H. carried out STM measurements and analysed the experimental data. S.H. and Y.M. carried out model calculations. T.N. and R.A. carried out LDOS calculations. Y.Y, C.J.B., S.H., T.N., T.H., and S.S. wrote the manuscript with inputs from all the authors.

## Competing interests

The authors declare no competing interests.
