## [Peer Review File · Nature Communications]

Reviewers' Comments:

Reviewer #1:

Remarks to the Author:

The work of the authors presents a scanning tunnelling spectroscopy study of complex magnetism in GdRu₂Si₂ films, which further confirms the emergence of multiple complex spin states (in addition to the preceding Nature Nanotechnology article by the authors) in this material. The experimental findings concerning the measured spectra are supported by theoretical calculations of various kind. The manuscript is potentially interesting, however at this point I cannot recommend the manuscript in its current shape for publication in Nature Communications, as I am not entirely convinced by the novelty aspect of this work, and I think that the focus of the manuscript has to be shifted more to the detailed discussion of the combined theory/experiment interpretation of the data. I elaborate in more detail below.

* I am not sure that I was able to catch the exact difference between the experimental technique of the authors, that they refer to as "spectroscopic imaging scanning tunneling microscopy", and the conventional spin-polarized or non-spin-polarized STM.

* In my view, the authors stress a bit too much the novelty of the following aspects:
(i) that the skyrmions, even very small, can be detected using non-magnetic probes, and
(ii) that the observed coupling between itinerant and localized moments can be detected by non-spin-polarized scanning tunnelling techniques.

Concerning (i), there is sizeable published effort e.g. from the Wiesendanger group, which in the course of the past decade discusses and employs non-magnetic STM for detection of various magnetic states, including skyrmions. I am not sure the authors really want to stress this aspect as one of the main points.

Concerning (ii), there is also plenty of existing experience in the STM community. STM as a technique is sensitive to electronic states which reach relatively far into the vacuum. Thus, s-, p- and d_{z²} states often serve as the "linking" states which lend themselves to the detection by tunnelling techniques, while at the same time carrying the information about the properties governed by more localized states, e.g. d and f.

Coming from the atomistic description, it is quite understandable and even expected that the spin-polarization of the localised f-states is also reflected in the modulation of the properties of itinerant e.g. s-states, which can then be in turn used to read off the magnetic contrast. As far as I am aware there are quite a few examples of this type in much less "exotic" systems in the STM literature, and I could refer here again to the works by Wiesendanger group.

Therefore, the fact that a scanning technique detects magnetic order mainly attributed to the localised states via measuring itinerant states is not perceived by me as truly surprising at this point. In this context the statement that "these observations were performed in systems where itinerant electrons themselves carry magnetic moments" strikes me as somewhat strange, since e.g. also in the model of the authors it is the fact that the conduction electrons carry a spin degree of freedom that gives rise to the observed modulation of the charge density, which would not be present given spinless electrons. Perhaps the authors could convince me that I am not interpreting these aspects properly.

In this respect, it is not clear which itinerant states the authors refer to, and I think some DFT calculations could really help here. The reference to the bulk calculations is not very helpful, as the situation at the surface can be drastically different from that in the bulk. In the current approach of the authors, the LDOS of the surface layer is analyzed,

and based on these conclusions on specific termination are made, however, to my knowledge, it is the LDOS in the vacuum (at a certain distance from the surface layer) which has to be assessed and compared to the experimental spectra so that a certain statement can be made. Of course in the Wannier function based approach of the authors LDOS in the vacuum is not accessible, moreover, I am not sure a simple termination of the Wannier Hamiltonian is a good approximation for addressing the surface properties on a 200 meV scale. I would encourage the authors to do a slab DFT calculation in order to learn more about the states which are responsible for the STM signal.

* I would also encourage the authors to put more focus on the analysis and discussion of their findings concerning the "clue to understand the skyrmion formation mechanism" in the studied material. Many of the details are hidden in the Methods section of the manuscript, while in my opinion they belong to the main text, which makes understanding of the result very difficult, especially the part concerning the analysis of emergent spin textures. The discussion part here constitutes only 2-3 paragraphs. Conceptually, leaving the aspect of coupling between localised and itinerant electrons aside, what is the main novelty of the findings as compared to the previous work by the authors presented in Refs. [9,13,14,20,38]?

* How do the contrast maps of Fig. 4 depend on the underlying electronic structure, e.g. in terms of band filling, hopping strength and strength of the exchange coupling?

Reviewer #2:

Remarks to the Author:

The main finding of this work is that there is a strong link between the itinerant electrons imaged in tunnelling experiments, and localised moments that form (skyrmion) spin structures. Due to this coupling, the charge sector shows characteristics of the spin sector, which is an interesting observation that can be very beneficial in the study of these materials. The good qualitative agreement between the experimental data, model calculations and previous field dependent studies make a strong case for the authors conclusions. Furthermore, from a technical point this is an impressive body of work; it is challenging to measure the magnetic field dependence on the same atomic location for such a large number of external magnetic fields. I therefore recommend publication of this work.

However, there are a few points that I would like the authors to address.

It is unclear to me why the authors present maps taken at -20mV, yet analyse the field dependence of their spectra at -70mV. Is the field dependence very different for these two voltages? Fig. 3i does indeed give this impression, but this may be affected by the setup-effect. Is there a reason the authors show normalised maps, yet regular spectra?

This makes me further wonder if there is also a voltage dependence of the real space patterns. Naively I would expect that if the LDOS patterns directly reflect the (static) spin structure, there should not be a bias voltage dependence. If, on the other hand, there is a bias voltage dependence, does the conclusion of the authors still hold? I hope the authors can comment on this.

The two topographies shown in Fig. 1c-d remind me a lot of the two predominant surface appearances observed on a related compound, URu₂Si₂. On that system, however, there was controversy about the assignment of the atomically resolved layer, as opposite conclusions were drawn from step height analysis on the one hand, and substitutional defects on the other. I was therefore wondering if the authors have found step edges between the two surface terminations they find on GdRu₂Si₂, and can perhaps use these to corroborate their surface assignment?

Comment on references:

Ref. 8: article number is 5831

Ref. 20: 'priprint' should be 'preprint'

Reviewer #3:

Remarks to the Author:

Y. Yasui and coworkers investigate GdRu_2Si_2 single crystal surfaces with non-magnetic STM and STS at $T=1.5$ Kelvin in a magnetic field regime of 0-12 Tesla. They find two different types of surfaces which they attribute to Si and Gd termination. The field-dependent data reveal different phases which are discriminated by different patterns in $dI/dV/(I/V)$ maps at low bias ($V=-20$ mV), by field-dependent laterally averaged dI/dV spectra and by the dI/dV signal at $V=-70$ mV. Building on a very recent publication, Ref. 9, these phases are identified as phase I: modulated helical spin spiral, phase II: square skyrmion lattice (in the field range $\sim 2.1-2.6$ T), phase III: 2Q fan-like structure, and phase IV: FM state. Supporting the measurements, the authors perform DFT calculations in the FM state and derive at the surface LDOS after a mapping to a tight binding model. In addition, atomistic spin MC calculations are performed, where the parameters (nearest-neighbor exchange, biquadratic term, anisotropy, zeeman energy) seem to be chosen in a way to qualitatively reproduce the expected magnetic phases, and these spin textures are then used to calculate charge distributions, to be compared to the spin-averaged experimental data.

In general, the experimental data is of very high quality and the manuscript is clearly written. It is debatable whether the presented findings establish a "new scheme" or rather extend an established scheme--exploiting LDOS variations to detect spin textures--to a new class of materials. Though the aim of the paper is well-chosen and timely, unfortunately, the manuscript is not entirely convincing in its present form, which I will try to explain point by point in the following.

1. **Determination of the surface termination**:

It seems that in Fig. 1(e,f) the experimental dI/dV spectra, which in good approximation represent the vacuum LDOS above the surface, are compared to the LDOS calculated at the atomic sites of the surface atoms. If this is correct, the similarity of spectra alone does not supply a strong argument for surface identification, because the LDOS can change drastically with distance due to different decay lengths of the contributing states. Maybe the authors can give additional arguments for their surface termination assignment.

2. **Normalization of dI/dV with I/V** in Fig. 2b and Fig.3a-d:

I understand that this procedure is common in the high bias regime (1V and above), e.g. when molecular states are investigated. I doubt however that this kind of normalization is a good procedure close to the Fermi-level ($V=-20$ mV), where I expect artifacts and an enhanced noise level due to vanishing values in the denominator. Instead, if the feedback circuit is an issue, dI/dV maps can be taken at constant height with the feedback switched off. Alternatively, at this low bias (-20 mV) LDOS variations should be directly visible in the z channel when the feedback is on.

3. **Low resolution in Fig.3(a-d)**:

Do I understand correctly that for the data in Fig. 3(a-d) full $dI/dV(V)$ spectra were taken at every pixel of the image? It seems that the image resolution is chosen too low to resolve all details of the lateral LDOS patterns. dI/dV maps at specific bias voltages might have been a better choice here. The diagonally running stripes in Fig.3a remain unexplained, unfortunately, but this seems to be admitted by the authors by stating that the Q_1+Q_2 spot in phase I is not explained.

4. **Energy dependence of the observed (NMR?) effect**:

The field-dependent spectroscopy shown in Fig. 3(i,j) seems to suggest that the impact of the spin texture onto the vacuum LDOS is strongest for the occupied states, but the energy dependence is not discussed in detail. An obvious question is, whether the effect is related or restricted to the entire LDOS peak at $V \sim -200$ mV and why this is the case.

5. **Lateral contrast and periods in phase II**:

After a close look at Fig. 4b, I do agree with the authors, that for this spin structure the magnetic pattern and the LDOS variations should have the same periods. It seems that the Zeeman term lifts the degeneracy between "up" and "down" skyrmions. In any case, since the atomic lattice is nicely resolved by the authors, the Q1 and Q2 values can be determined very accurately. How do they compare to the 0.22 determined in Ref. 9? Is the spin texture in phase II commensurate or practically incommensurate with the atomic lattice? Are there defects or distortions seen in the magnetic lattice or a specific type of coupling to defects or step edges?

6. **Local spectroscopy**:

The authors compare laterally averaged spectra at different magnetic field values, but not local spectra at different sites at fixed field values, e.g. spectra measured above an "up" and a "down" skyrmion in phase II. The data seems to be already available and the contrast is sizable. More might be learned about the NCMR effect in this system by laterally resolved dI/dU spectra.

7. **Reproducibility**:

Accidental tip changes can happen, especially in long measurement series. There are some indications that surface defects vanished or changed within the field-dependent measurements in Fig. 3b (lower left) and supplementary Fig. 2 from c to d. Why are there no defects seen at all at 12 Tesla, except for two indentations where 2 protrusions were seen at 3 Tesla? As an additional complication, a tip might become spin-sensitive unintentionally by picking up magnetic material from the surface, see e.g. S. Loth et al., Science 335, 196 (2012). Due to these general difficulties of STM, I would like to know to what extent the presented data were reproduced with different tips in different measurement series. Are the same LDOS patterns observed when the magnetic field is swept down from 12 Tesla? To what extent can the curve in Fig. 3k be reproduced in a downward sweep or in a second upward sweep of the magnetic field?

8. **Details of the spin textures**:

Reference 9, where the different phases, their symmetries and periods and critical fields were already determined, supplies a good starting point for this investigation. As far as I understand, however, the spin textures were not unambiguously determined in all detail in Ref. 9, e.g. because Lorentz TEM is not sensitive to out-of-plane spin components. This is not an ideal situation to explore a new technique or extend an established technique to a new class of material; are there spin-polarized STM measurements available for GdRu_2Si_2 to support the interpretation of the spin-averaged data?

9. **Relation of this work to the NCMR effect in Ref. 27**:

On page 10 the authors state: "Since the itinerant electrons' spins are aligned with localised moments, kinetic energy of the itinerant electrons depends on the *relative angle between localised magnetic moments* at neighbouring sites. Thus, itinerant electrons reflect local magnetic structures. ". The authors seem to suggest that the observed effect is similar to the NCMR effect observed in 3d transition metal systems. Can a TAMR like origin (Ref. 24) be ruled out for the data on GdRu_2Si_2 or are there at least arguments why such a scenario is less likely? And second, should one view the proposed model to describe the LDOS variations, where itinerant electrons are exchange coupled to local spins, as a variant of the NCMR effect in local moment systems, or rather as a microscopically distinct effect which effectively scales like the NCMR effect in 3d transition metal systems?

To Reviewer #1

1. *The work of the authors presents a scanning tunnelling spectroscopy study of complex magnetism in $GdRu_2Si_2$ films, which further confirms the emergence of multiple complex spin states (in addition to the preceding Nature Nanotechnology article by the authors) in this material. The experimental findings concerning the measured spectra are supported by theoretical calculations of various kind. The manuscript is potentially interesting, however at this point I cannot recommend the manuscript in its current shape for publication in Nature Communications, as I am not entirely convinced by the novelty aspect of this work, and I think that the focus of the manuscript has to be shifted more to the detailed discussion of the combined theory/experiment interpretation of the data. I elaborate in more detail below.*

[Response]

We are pleased that the reviewer evaluated our work by stating “*The manuscript is potentially interesting.*”

Following the reviewer’s suggestion, we revised the manuscript to put more focus on the discussion of the interpretation of the data, which will lead to elucidating the formation mechanism of the new type of square skyrmion lattice in the centrosymmetric crystal structure.

2. **I am not sure that I was able to catch the exact difference between the experimental technique of the authors, that they refer to as “spectroscopic imaging scanning tunneling microscopy”, and the conventional spin-polarized or non-spin-polarized STM.*

[Response]

The term “spectroscopic-imaging scanning tunneling microscopy” (SI-STM), refers to the technique of collecting both $I(V)$ and $dI(V)/dV$ spectra at each pixel of the usual constant-current topography image, forming the 3D data sets $I(x, y, V)$ and $dI(x, y, V)/dV$. We added this explanation for SI-STM in the main text (line 78). A given layer of the $dI(x, y, V)/dV$ data set at a constant bias, say $dI(x, y, V = 20 \text{ meV})/dV$ represents an image of the spectroscopic features present at the corresponding energy (upon a suitable normalization procedure). Note that this technique can be used either with or without a spin-polarized tip. In our case we employ SI-STM with a non-spin-polarized tip, in which case the images correspond closely to the spin-averaged density of states.

SI-STM is different from topography, where the density of states (DOS) information between the Fermi level and the setup sample bias voltage is summed up. SI-STM is also different from dI/dV maps at a single bias voltage because it can investigate the bias dependence and spatial modulations simultaneously, and it structures the data in a form amenable to normalization which compensates for setpoint-induced artifacts.

3. **In my view, the authors stress a bit too much the novelty of the following aspects: (i) that the skyrmions, even very small, can be detected using non-magnetic probes, and (ii)*

that the observed coupling between itinerant and localized moments can be detected by non-spin-polarized scanning tunnelling techniques.

Concerning (i), there is sizeable published effort e.g. from the Wiesendanger group, which in the course of the past decade discusses and employs non-magnetic STM for detection of various magnetic states, including skyrmions. I am not sure the authors really want to stress this aspect as one of the main points.

[Response]

We agree that we emphasized too much on the detection scheme, and we do not attempt to advance this as a main point of this manuscript. We revised the abstract and introductory part (line 49 – 58) to clarify this point.

4. *Concerning (ii), there is also plenty of existing experience in the STM community. STM as a technique is sensitive to electronic states which reach relatively far into the vacuum. Thus, s-, p- and d_{z^2} states often serve as the “linking” states which lend themselves to the detection by tunnelling techniques, while at the same time carrying the information about the properties governed by more localized states, e.g. d and f.*

Coming from the atomistic description, it is quite understandable and even expected that the spin-polarization of the localised f-states is also reflected in the modulation of the properties of itinerant e.g. s-states, which can then be in turn used to read off the magnetic contrast. As far as I am aware there are quite a few examples of this type in much less “exotic” systems in the STM literature, and I could refer here again to the works by Wiesendanger group.

Therefore, the fact that a scanning technique detects magnetic order mainly attributed to the localised states via measuring itinerant states is not perceived by me as truly surprising at this point. In this context the statement that “these observations were performed in systems where itinerant electrons themselves carry magnetic moments” strikes me as somewhat strange, since e.g. also in the model of the authors it is the fact that the conduction electrons carry a spin degree of freedom that gives rise to the observed modulation of the charge density, which would not be present given spinless electrons. Perhaps the authors could convince me that I am not interpreting these aspects properly.

[Response]

We apologize that our writing of “systems where itinerant electrons themselves carry magnetic moments” was not clear. As the reviewer pointed out, the spin degree of freedom of itinerant electrons is also relevant in our model, but there is an important difference between the previous experiments and our work.

The previous experiments that the reviewer mentions have been performed in 3d transition metals where the magnetic structures (e.g. skyrmions) are formed with the magnetic moments carried by itinerant electrons. This leads to the spatial modulation in

the density of states (DOS) via tunneling anisotropic magnetoresistance (TAMR) and/or non-collinear magnetoresistance (NCOMR) effects.

In the case of GdRu_2Si_2 , by contrast, the magnetic moments of itinerant electrons do not form magnetic structures by themselves. This is evidenced by the fact that the sister compound LaRu_2Si_2 without f electrons is nonmagnetic [JPSJ **84**, 063702 (2015)]. All the magnetic structures are associated with the localized Gd $4f$ states, which have negligible DOS near E_F . Since our STM experiments have been performed near E_F , energetically far ($\sim\text{eV}$) from the magnetic Gd $4f$ states, the observation of the magnetic structures in the DOS near E_F is non-trivial and worthy of consideration. To the best of our knowledge, there is no other work that investigated localized f -electron states using STM.

We believe that our results have twofold importance. One is that magnetic structures have been detected with non-spin-polarized STM even in a system where the magnetic moments are localized. Nevertheless, we recognize that we emphasized this aspect too much in the previous manuscript.

In the revised manuscript, we have amended the text so as to further highlight the other point, namely the clue to the skyrmion formation mechanism. The key ingredient in our model is the exchange coupling between itinerant electron's spin and localized moments (c - f coupling). It has been guessed that magnetic orders including the square skyrmion lattice in GdRu_2Si_2 may be mediated through the c - f coupling but no experimental evidence had been obtained. Our combined experimental/theoretical work shows that the qualitative features (e.g. Q components) of the DOS modulations can be nicely reproduced by introducing the c - f coupling. This evidences that the c - f coupling does play an important role in GdRu_2Si_2 and sheds light on the microscopic model of the skyrmion formation mechanism.

- In this respect, it is not clear which itinerant states the authors refer to, and I think some DFT calculations could really help here. The reference to the bulk calculations is not very helpful, as the situation at the surface can be drastically different from that in the bulk. In the current approach of the authors, the LDOS of the surface layer is analyzed, and based on this conclusions on specific termination are made, however, to my knowledge, it is the LDOS in the vacuum (at a certain distance from the surface layer) which has to be assessed and compared to the experimental spectra so that a certain statement can be made. Of course in the Wannier function based approach of the authors LDOS in the vacuum is not accessible, moreover, I am not sure a simple termination of the Wannier Hamiltonian is a good approximation for addressing the surface properties on a 200 meV scale. I would encourage the authors to do a slab DFT calculation in order to learn more about the states which are responsible for the STM signal.*

[Response]

A similar suggestion was also given by Reviewer #3. Following the reviewers' suggestion, we performed slab DFT calculations for vacuum LDOS 5 Å above the surface and put the new results to Figs. 1e and 1f. These results reproduced the observed features even better. We appreciate the reviewer for making us realize this.

Figure 1: Comparison of measured dI/dV curves with newly calculated vacuum LDOS.

As for the relevant itinerant states, we would like to note that our model includes the exchange coupling between itinerant electrons and localized moments in a general form (Eq. 2) and explains the observed Q components in each magnetic phase qualitatively. Therefore, we judge that our arguments hold regardless of the orbital characters of itinerant states.

We are aware that there remain features that are not captured in our model (e.g. $Q_1 + Q_2$ modulation in Phase I, difference between Si and Gd terminated surfaces), and agree with the reviewer that identifying the exact orbital characters may be important to settle these future issues. Nevertheless, we believe that our minimal model is a reasonable starting point and helps to understand the physics of GdRu_2Si_2 .

6. **I would also encourage the authors to put more focus on the analysis and discussion of their findings concerning the “clue to understand the skyrmion formation mechanism” in the studied material. Many of the details are hidden in the Methods section of the manuscript, while in my opinion they belong to the main text, which makes understanding of the result very difficult, especially the part concerning the analysis of emergent spin textures. The discussion part here constitutes only 2-3 paragraphs. Conceptually, leaving the aspect of coupling between localised and itinerant electrons aside, what is the main novelty of the findings as compared to the previous work by the authors presented in Refs. [9,13,14,20,38]?*

[Response]

Related to the 4th point, we agree to put more focus on the clue to understand the skyrmion formation mechanism. We therefore moved the section of “Calculation of magnetic configuration and charge density” from methods to the main text (line 128 – 167) as suggested.

The novelty in the model, compared to refs. 9, 13, 14, 20, 38, is the anisotropy due to the symmetry of the tetragonal crystal structure. This anisotropy enables the square

skyrmion lattice in a square crystal structure. Please note that only triangular skyrmion lattices in triangular crystal structures were expected in the previous theories, in which the anisotropy was not considered. We added this description in the main text (line 144 – 146).

7. **How do the contrast maps of Fig. 4 depend on the underlying electronic structure, e.g. in terms of band filling, hopping strength and strength of the exchange coupling?*

[Response]

We added figures of charge density modulations calculated for different chemical potentials as Supplementary Figure 11. The Q -components are robust against the change of chemical potential, which corresponds to the band filling. Since the Q components in the charge density reflect the magnetic structures of the localized moments, which are pre-determined with the effective spin model, it is unlikely that parameters for the itinerant electrons alter the Q components.

To Reviewer #2

1. *The main finding of this work is that there is a strong link between the itinerant electrons imaged in tunnelling experiments, and localised moments that form (skyrmion) spin structures. Due to this coupling, the charge sector shows characteristics of the spin sector, which is an interesting observation that can be very beneficial in the study of these materials. The good qualitative agreement between the experimental data, model calculations and previous field dependent studies make a strong case for the authors conclusions. Furthermore, from a technical point this is an impressive body of work; it is challenging to measure the magnetic field dependence on the same atomic location for such a large number of external magnetic fields. I therefore recommend publication of this work.*

However, there are a few points that I would like the authors to address.

[Response]

We are pleased that the reviewer evaluated our manuscript by stating “*I therefore recommend publication of this work.*”

2. *It is unclear to me why the authors present maps taken at -20mV, yet analyse the field dependence of their spectra at -70mV. Is the field dependence very different for these two voltages? Fig. 3i does indeed give this impression, but this may be affected by the setup-effect. Is there a reason the authors show normalised maps, yet regular spectra?*

[Response]

We showed the values at -70 mV in Fig. 3k because the transitions are clearest at this voltage. This figure is used to illustrate the transitions between magnetic phases especially to clarify the first-order like and second-order like transitions. We confirmed that the field dependence is similar at any bias voltages including -20 mV; first-order like discontinuous jump across Phase II, and second-order like gradual change from Phase III to the fully polarized phase. We added field dependence of dI/dV at -20 mV in Supplementary Information (Supplementary Figure 8).

The same normalization that we have used to display meaningful spectroscopic images at a particular bias, here -20 mV, renders it nontrivial to make comparisons between different biases, e.g. within a single spectroscopy curve. While the $dI(V)/dV$ curve has a clear physical meaning, that is approximately proportional to the density of states, the low-energy features tend to be emphasized in the normalized spectrum because $[dI(V)/dV]/[I(V)/V] = d[\log I(V)]/d[\log V]$. We also added magnetic field dependence of the normalized conductance L (corresponds to Figs. 3i and j) and a cut of L at -70 mV (corresponds to Fig. 3k) as Supplementary Figure 10.

3. *This makes me further wonder if there is also a voltage dependence of the real space patterns. Naively I would expect that if the LDOS patterns directly reflect the (static) spin structure, there should not be a bias voltage dependence. If, on the other hand,*

there is a bias voltage dependence, does the conclusion of the authors still hold? I hope the authors can comment on this.

[Response]

The spatial patterns of L maps appear to be somewhat different depending on the bias voltage because the relative intensity/sign of Q components vary with bias voltage. However, their Fourier components do not show bias dependence except in their intensities. This is consistent with the picture that the Q -structures of the density of states are imprinted from the static magnetic structures. We added L maps at different slices of bias voltage to Supplementary Information (Supplementary Figure 4).

4. *The two topographies shown in Fig. 1c-d remind me a lot of the two predominant surface appearances observed on a related compound, URu₂Si₂. On that system, however, there was controversy about the assignment of the atomically resolved layer, as opposite conclusions were drawn from step height analysis on the one hand, and substitutional defects on the other. I was therefore wondering if the authors have found step edges between the two surface terminations they find on GdRu₂Si₂, and can perhaps use these to corroborate their surface assignment?*

[Response]

For the step-height analysis similar to that in [P. Aynajian *et al.*, PNAS **107**, 10383 (2010)], we need a sequence of mono-layer steps. However, we did not obtain steps whose height corresponds to a mono-layer thickness (approximately a quarter of the unit cell height), and hence we cannot make this analysis.

Besides, the apparent step height between different terminations measured with STM does not necessarily equal the height between center of the mass of atoms. If, for example, the LDOS and/or the work functions are different between two terminations, the apparent step height depends on the parameters for the feedback loop (setup condition). This could even invert the magnitude relationship of the apparent step height between two different terminations. Thus, we do not rely on step-height analysis even if available.

For the substitutional defects used in [A. R. Schmidt *et al.*, Nature **465**, 570 (2010)], only stoichiometric samples are available for GdRu₂Si₂ so far, and this method is not possible at this time.

5. *Comment on references:*
Ref. 8: article number is 5831
Ref. 20: 'priprint' should be 'preprint'

[Response]

We appreciate for pointing out these mistakes. We revised the references 8 and 20.

To Reviewer #3

1. *Y. Yasui and coworkers investigate GdRu₂Si₂ single crystal surfaces with non-magnetic STM and STS at T = 1.5 Kelvin in a magnetic field regime of 0-12 Tesla. They find two different types of surfaces which they attribute to Si and Gd termination. The field-dependent data reveal different phases which are discriminated by different patterns in dI/dV/(I/V) maps at low bias (V=-20 mV), by field-dependent laterally averaged dI/dV spectra and by the dI/dV signal at V=-70mV. Building on a very recent publication, Ref. 9, these phases are identified as phase I: modulated helical spin spiral, phase II: square skyrmion lattice (in the field range 2.1-2.6T), phase III: 2Q fan-like structure, and phase IV: FM state. Supporting the measurements, the authors perform DFT calculations in the FM state and derive at the surface LDOS after a mapping to a tight binding model. In addition, atomistic spin MC calculations are performed, where the parameters (nearest-neighbor exchange, biquadratic term, anisotropy, zee-man energy) seem to be chosen in a way to qualitatively reproduce the expected magnetic phases, and these spin textures are then used to calculate charge distributions, to be compared to the spin-averaged experimental data. In general, the experimental data is of very high quality and the manuscript is clearly written. It is debatable whether the presented findings establish a "new scheme" or rather extend an established scheme—exploiting LDOS variations to detect spin textures—to a new class of materials. Though the aim of the paper is well-chosen and timely, unfortunately, the manuscript is not entirely convincing in its present form, which I will try to explain point by point in the following.*

[Response]

We are pleased that the reviewer evaluated our manuscript by stating “*the experimental data is of very high quality and the manuscript is clearly written*”

We put too much emphasis on the detection scheme in the previous manuscript. We therefore revised the manuscript to clarify that the main focus of the manuscript is not to show the detection scheme but to discuss the formation mechanism of the new type of square skyrmion lattice in the centrosymmetric crystal GdRu₂Si₂.

2. *Determination of the surface termination: It seems that in Fig. 1(e,f) the experimental dI/dV spectra, which in good approximation represent the vacuum LDOS above the surface, are compared to the LDOS calculated at the atomic sites of the surface atoms. If this is correct, the similarity of spectra alone does not supply a strong argument for surface identification, because the LDOS can change drastically with distance due to different decay lengths of the contributing states. Maybe the authors can give additional arguments for their surface termination assignment.*

[Response]

Similar suggestion was also given by Reviewer #1. Following the reviewers' suggestion, we performed slab DFT calculations for vacuum LDOS 5 Å above the surface,

and the results reproduced the observed features even better (Figs. 1e and 1f). We appreciate the reviewer for making us realize this.

Figure 2: Comparison of measured dI/dV curves with newly calculated vacuum LDOS.

3. *Normalization of dI/dV with I/V in Fig. 2b and Fig.3a-d: I understand that this procedure is common in the high bias regime (1V and above), e.g. when molecular states are investigated. I doubt however that this kind of normalization is a good procedure close to the Fermi-level ($V=-20$ mV), where I expect artifacts and an enhanced noise level due to vanishing values in the denominator. Instead, if the feedback circuit is an issue, dI/dV maps can be taken at constant height with the feedback switched off. Alternatively, at this low bias (-20 mV) LDOS variations should be directly visible in the z channel when the feedback is on.*

[Response]

As the reviewer mentioned, the normalized conductance $L = \frac{dI}{dV} / \frac{I}{V}$ is used in the main text to suppress the set-point related artifacts that contaminate raw dI/dV maps. The most serious issue in the raw dI/dV map is that any LDOS modulation appearing between E_F and the set-up bias may appear even at different biases, giving rise to an extrinsic non-dispersive feature in the map [Science **315**, 1380 (2007)]. The L map is free from this issue and we can safely conclude that the non-dispersive features observed in this work are associated with the static magnetic orders.

The noise level at low biases is indeed enhanced, but Figs. 2 and 3 retain a high enough signal-to-noise ratio to conclude the formation of structures in the DOS. Nevertheless, we understand that it may be helpful for readers to know raw dI/dV maps as well. Therefore, we added raw conductance dI/dV maps in Supplementary Information (Supplementary Figure 6).

Unfortunately, dI/dV maps taken at constant height may include another artifact due to the spatial variation of the work function.

LDOS variations are directly observed in the z channel with feedback on for the Gd-terminated surface as shown in Fig. 1d. With this method, however, spectrum at each point cannot be obtained and the sensitivity to the LDOS modulations in the topographic image is logarithmically lower compared to the dI/dV or L maps. In summary, we judged that L map is the best practical choice for our purpose.

4. *Low resolution in Fig.3(a-d): Do I understand correctly that for the data in Fig. 3(a-d) full $dI/dV(V)$ spectra were taken at every pixel of the image? It seems that the image resolution is chosen too low to resolve all details of the lateral LDOS patterns. dI/dV maps at specific bias voltages might have been a better choice here. The diagonally running stripes in Fig.3a remain unexplained, unfortunately, but this seems to be admitted by the authors by stating that the $Q1+Q2$ spot in phase I is not explained.*

[Response]

As the reviewer stated, a full $I-V$ and dI/dV curves are measured at every pixel of the image in Figs. 3a-d. Each image is cropped from a larger 30×30 nm image, for which the pixel density was optimized to cover the 1st Brillouin zone in the Fourier-transformed map.

We have also measured a conductance map in Phase II with better spatial resolution and added it to Supplementary Information (Supplementary Figure 3). We confirmed no additional fine structures are there.

dI/dV maps at specific bias voltages would unavoidably suffer from artifacts due to the set-point effect. By using the SI-STM technique and the $\frac{dI}{dV}/\frac{I}{V}$ normalization procedure, we can obtain more accurate LDOS images at a certain bias voltage.

As pointed out, the diagonally running stripes in Phase I corresponds to the $Q_1 + Q_2$ modulation, and we do not have explanation within our model.

5. *Energy dependence of the observed (NCMR?) effect: The field-dependent spectroscopy shown in Fig. 3(i,j) seems to suggest that the impact of the spin texture onto the vacuum LDOS is strongest for the occupied states, but the energy dependence is not discussed in detail. An obvious question is, whether the effect is related or restricted to the entire LDOS peak at $V = -200$ mV and why this is the case.*

[Response]

The effect of magnetic order is not restricted to the LDOS peak at -200 mV because the LDOS shows modulations at least within ± 100 mV as shown in the newly added Supplementary Figure 4. Details of the magnetic-field effect at different energies remain for future investigations.

6. *Lateral contrast and periods in phase II: After a close look at Fig. 4b, I do agree with the authors, that for this spin structure the magnetic pattern and the LDOS variations should have the same periods. It seems that the Zeeman term lifts the degeneracy between "up" and "down" skyrmions. In any case, since the atomic lattice is nicely resolved by the authors, the $Q1$ and $Q2$ values can be determined very accurately. How do they compare to the 0.22 determined in Ref. 9? Is the spin texture in phase II commensurate or practically incommensurate with the atomic lattice? Are there defects or distortions seen in the magnetic lattice or a specific type of coupling to defects or step edges?*

[Response]

From our results, the length of the Q -vector can be determined up to two digits ($|Q| = 0.22$). This precision is limited mainly by the field of view of $30 \times 30 \text{ nm}^2$. With this precision, we cannot tell if the magnetic structures are perfectly incommensurate or commensurate with a large unit cell, for example $2/9 = 0.222\dots$

We do not observe any distortion in magnetic structure at surface impurities and step edges. We added Supplementary Figure 3 to show a slide of magnetic structure during the measurement. This suggests that the magnetic structures are not pinned to the surface impurities.

7. *Local spectroscopy: The authors compare laterally averaged spectra at different magnetic field values, but not local spectra at different sites at fixed field values, e.g. spectra measured above an "up" and a "down" skyrmion in phase II. The data seems to be already available and the contrast is sizable. More might be learned about the NCMR effect in this system by laterally resolved dI/dU spectra.*

[Response]

We added dI/dV curves at different positions in Supplementary Information (Supplementary Figure 5). The change in spectra is most significant near zero bias.

8. *Reproducibility: Accidental tip changes can happen, especially in long measurement series. There are some indications that surface defects vanished or changed within the field-dependent measurements in Fig. 3b (lower left) and supplementary Fig. 2 from c to d. Why are there no defects seen at all at 12 Tesla, except for two indentations where 2 protrusions were seen at 3 Tesla? As an additional complication, a tip might become spin-sensitive unintentionally by picking up magnetic material from the surface, see e.g. S. Loth et al., Science 335, 196 (2012). Due to these general difficulties of STM, I would like to know to what extent the presented data were reproduced with different tips in different measurement series. Are the same LDOS patterns observed when the magnetic field is swept down from 12 Tesla? To what extent can the curve in Fig. 3k be reproduced in a downward sweep or in a second upward sweep of the magnetic field?*

[Response]

We added topographic images simultaneously measured during dI/dV maps as Supplementary Figure 7. As shown in the topographic images, the appearance of the impurities does not change at all for different magnetic fields, ensuring that the shape of the tip apex did not change. The impurities look differently in the L maps because the DOS of impurities relative to GdRu_2Si_2 apparently changes depending on the magnetic field. Such magnetic-field induced change in the LDOS can be canceled out easily in the topographic image taken at a higher bias.

In order to further show that our tip was intact throughout the measurement, we also attach below the same topographic images but without subtraction of background.

Besides the smooth evolution associated with the piezo scanner creep, the tip-sample distance did not have any discontinuous change.

Figure 3: Raw topographic images simultaneously measured with Fig. 3a-d of the main text. $V_s = 100$ mV and $I_s = 100$ pA.

We used five scanning tips and seven GdRu_2Si_2 samples in total. We confirmed all these sets gives similar spectra.

If the scanning tip were to be magnetized due to unintentional pick-up of Gd atoms, a \mathbf{Q}_1 structure would have been observed in Phase I because the scanning tip is now sensitive to the direction of magnetic moments. Therefore, we conclude that the measurements were performed with non-magnetic scanning tips. We described this in the main text (line 126).

The curve in Fig. 3k is very reproducible when magnetic field is swept down from 12 T, except slight shifts in the transition field values across the Phase II due to their first-order nature. We added corresponding figure in Supplementary Information (Supplementary Figure 9).

9. *Details of the spin textures: Reference 9, where the different phases, their symmetries and periods and critical fields where already determined, supplies a good starting point for this investigation. As far as I understand, however, the spin textures were not unambiguously determined in all detail in Ref. 9, e.g. because Lorentz TEM is not sensitive to out-of-plane spin components. This is not an ideal situation to explore a new technique or extent an established technique to a new class of material; are there spin-polarized STM measurements available for GdRu_2Si_2 to support the interpretation of the spin-averaged data?*

[Response]

It is correct that the magnetic structures are not unambiguously determined in ref. 9. Spin-polarised measurements are not available with our current experimental setup, and we do not know of any STM measurements with or without spin sensitivity reported by any other group.

10. *Relation of this work to the NCMR effect in Ref. 27: On page 10 the authors state: “Since the itinerant electrons’ spins are aligned with localised moments, kinetic energy of the itinerant electrons depends on the relative angle between localised magnetic moments at neighbouring sites. Thus, itinerant electrons reflect local magnetic*

structures.” The authors seem to suggest that the observed effect is similar to the NCMR effect observed in 3d transition metal systems. Can a TAMR like origin (Ref. 24) be ruled out for the data on GdRu₂Si₂ or are there at least arguments why such a scenario is less likely? And second, should one view the proposed model to describe the LDOS variations, where itinerant electrons are exchange coupled to local spins, as a variant of the NCMR effect in local moment systems, or rather as a microscopically distinct effect which effectively scales like the NCMR effect in 3d transition metal systems?

[Response]

Even if a significant TAMR effect was present, it would not suffice to explain the current observation. This is because the centre and the edges of skyrmions show different contrast, whereas spins pointing in and out of the surface should appear similarly for TAMR effect. We added description on this in the main text (line 184 – 186).

As the reviewer correctly pointed out, the phenomenon we observed can be regarded as a variant of the NCMR effect. However, it is different in the sense that coupling between itinerant electrons and localized magnetic moments are mandatory to observe the present LDOS modulations. Therefore, we can use the present results to establish the existence of the coupling, which has not been experimentally reported.

The observed coupling between itinerant electrons and localized magnetic moments not only enables us to access the localized f electrons using STM but also may help to stabilize the new type of square skyrmion lattice. We found that the previous manuscript put too much emphasis on the detection of magnetic structure using non-magnetic tips. We revised the manuscript to highlight that the main point of the manuscript is to shed light on the formation mechanism of the square skyrmion lattice in the centrosymmetric crystal GdRu₂Si₂ (line 49 – 58).

List of Changes

The changes are highlighted with color in the attached file “corrections.pdf”. Please refer to this file as well.

abstract: we moved the references to the first paragraph.

line 49 – 58: We revised the description to put more focus on the skyrmion formation mechanism instead of a detection scheme.

line 69: We revised the description according to the change in DFT calculation method.

line 78: We added a sentence to explain the spectroscopic-imaging STM.

Figure 1e and f: DFT calculation is updated for slab systems.

lines 91, 96, 114, and 157: We added references to Supplementary Figures.

line 126: We added a sentence to decline a possible magnetization of scanning tip.

line 128 – 167: We moved section “Calculation of magnetic configuration and charge density” from methods to the main text.

line 183 – 191: We revised the description for the relation to TAMR and NCMR effects.

line 198 – 203: We revised the conclusion part to clarify the main focus of the manuscript is to shed light on the formation mechanism of the square skyrmion lattice in the centrosymmetric crystal.

Methods: We revised the description according to the change in DFT calculation method.

Supplementary Note 2: We added this section for additional data (Supplementary Figures 3-11) to support the main observation.

Others: Several small modifications are made, mainly to reflect changes in the structure of the manuscript. These changes are also highlighted in colors.

Reviewers' Comments:

Reviewer #1:

Remarks to the Author:

I have carefully considered the response of the authors to my comments, as well as to the comments by the other referees. I appreciate the work that the authors have done in shifting the focus of their work to the discussion of the texture formation mechanisms. The findings will probably have a strong impact on the development of the field into the area of non-collinear magnetism in f-electron systems. In my view, the data are of a very high quality and the interpretation is believable enough to grant the publication in Nature Communications.

Reviewer #2:

Remarks to the Author:

The authors have sufficiently addressed my comments and questions, as well as those by the other referees, and have modified the main text and supplementary information accordingly. I recommend publication in Nature Communications.

To Reviewer #1

I have carefully considered the response of the authors to my comments, as well as to the comments by the other referees. I appreciate the work that the authors have done in shifting the focus of their work to the discussion of the texture formation mechanisms. The findings will probably have a strong impact on the development of the field into the area of non-collinear magnetism in f-electron systems. In my view, the data are of a very high quality and the interpretation is believable enough to grant the publication in Nature Communications.

[Response]

We are pleased that the reviewer evaluated our manuscript by stating “*the data are of a very high quality and the interpretation is believable enough to grant the publication in Nature Communications.*”

To Reviewer #2

The authors have sufficiently addressed my comments and questions, as well as those by the other referees, and have modified the main text and supplementary information accordingly. I recommend publication in Nature Communications.

[Response]

We are pleased that the reviewer evaluated our manuscript by stating “*I recommend publication in Nature Communications.*”